# Nanotechnologies in Delivery of DNA and mRNA Vaccines to the Nasal and Pulmonary Mucosa

**DOI:** 10.3390/nano12020226

**Published:** 2022-01-11

**Authors:** Jie Tang, Larry Cai, Chuanfei Xu, Si Sun, Yuheng Liu, Joseph Rosenecker, Shan Guan

**Affiliations:** 1Department of Pediatrics, Ludwig-Maximilians University of Munich, 80337 Munich, Germany; j.tang3@uq.edu.au; 2Australian Institute for Bioengineering and Nanotechnology, The University of Queensland, Brisbane 4072, Australia; larry.cai@uq.edu.au; 3National Engineering Research Center of Immunological Products, Department of Microbiology and Biochemical Pharmacy, Third Military Medical University, Chongqing 400038, China; xu1982978434@163.com (C.X.); sunsi90@163.com (S.S.); yhliu2017@lzu.edu.cn (Y.L.)

**Keywords:** DNA vaccine, mRNA vaccine, mucosal immune response, intranasal delivery, pulmonary delivery, nanoparticles

## Abstract

Recent advancements in the field of in vitro transcribed mRNA (IVT-mRNA) vaccination have attracted considerable attention to such vaccination as a cutting-edge technique against infectious diseases including COVID-19 caused by SARS-CoV-2. While numerous pathogens infect the host through the respiratory mucosa, conventional parenterally administered vaccines are unable to induce protective immunity at mucosal surfaces. Mucosal immunization enables the induction of both mucosal and systemic immunity, efficiently removing pathogens from the mucosa before an infection occurs. Although respiratory mucosal vaccination is highly appealing, successful nasal or pulmonary delivery of nucleic acid-based vaccines is challenging because of several physical and biological barriers at the airway mucosal site, such as a variety of protective enzymes and mucociliary clearance, which remove exogenously inhaled substances. Hence, advanced nanotechnologies enabling delivery of DNA and IVT-mRNA to the nasal and pulmonary mucosa are urgently needed. Ideal nanocarriers for nucleic acid vaccines should be able to efficiently load and protect genetic payloads, overcome physical and biological barriers at the airway mucosal site, facilitate transfection in targeted epithelial or antigen-presenting cells, and incorporate adjuvants. In this review, we discuss recent developments in nucleic acid delivery systems that target airway mucosa for vaccination purposes.

## 1. Introduction

In recent decades, nanotechnologies for the production of high-quality nucleic acids and efficient in vivo delivery systems have revolutionized the field of vaccine development [1,2,3,4]. Lauded as “third-generation vaccines”, DNA and in vitro transcribed messenger RNA (IVT-mRNA) harbor huge potential to offer pioneering solutions for clinically unmet needs [5,6]. Instead of using inactivated or live, attenuated viruses synthesized through time-consuming and demanding production procedures, vaccine developers now adopt simpler, rationally designed DNA constructs or IVT-mRNA to instruct cells of the recipient to express antigenic proteins that imitate parts of the target bacteria or viruses in order to generate antigen-specific humoral and cellular immunity to eliminate the real pathogens containing said antigen when they invade the host. Since host cells are responsible for antigen production, correct folding of the protein and natural glycosylation are guaranteed [7]. However, successful transfer of these antigen-encoding genetic materials to the target site (i.e., the nucleus for DNA transcription or ribosomes in the cytoplasm for mRNA translation) within host cells is a prerequisite to achieve this goal. Apart from several exceptional cases [8,9,10], delivery vehicles or devices are imperative because of limited cellular uptake and the instability of naked DNA and IVT-mRNA [2,11,12,13]. Nucleic acid-based vaccines (NA-vaccines) utilize affordable and well-established standard manufacturing processes that can scale up rapidly in response to outbreaks of infectious diseases, avoiding the complex procedures of repeated culture and inactivation of infectious pathogens or purification of recombinant antigens [5,14]. Furthermore, these genetic vaccines offer several distinct advantages over conventional vaccines (Table 1). NA-vaccines are highly flexible, capable of encoding virtually any type of protein. The successful implementation of these vaccines would prevent or treat, at least theoretically, any disease with effective and well-characterized antigen targets. Indeed, several NA-vaccines encoding viral antigens have been tested in clinical trials to investigate their protective potency and immunogenicity [15,16,17,18,19,20]. The range of diseases that could be addressed by NA-vaccines even extends to cancer [21,22,23]. The emergence of severe acute respiratory syndrome coronavirus 2 (SARS-CoV-2) has turned the field’s spotlight towards the prevention of coronavirus disease 2019 (COVID-19). Since then, research institutions and companies have raced to develop COVID-19 vaccines [24]. IVT-mRNA vaccines turned out to be the “biggest winner” of this vaccine competition, as they not only enabled an unimaginable speed of vaccine development but displayed extremely high protection rates against COVID-19. After the viral genome sequence was publicly released, it took just sixty-six days for Moderna’s 1273-mRNA vaccine candidate to enter phase I clinical trial [25]. Both Moderna’s 1273-mRNA vaccine and BioNTech’s 162b2-mRNA vaccine finished phase III clinical trials with 94–95% effectiveness in protection against COVID-19 and obtained emergency use authorization within one year from the outbreak of the virus [26,27].

One key obstacle on the path of developing advanced vaccines relates to the vaccination route and mucosal immunity. Many pathogens, including SARS-CoV-2, infect the host through the respiratory mucosa, so the interaction between the virus and the immune system first occurs in the mucosa of the respiratory tract. The mucosal immune system plays a key role in the host’s resistance to pathogen invasion [28,29]. However, most clinical available vaccines employ invasive parenteral routes when inoculated, such as intramuscular, subcutaneous, and intradermal injections [30]. The modality of vaccine administration could have an enormous impact on the efficacy of a vaccine, as it affects the accessibility of immune cells for priming and the extent of consequential effects (systemic and local immune responses) after the vaccination. In addition to a few reports using retinoic acid as an adjuvant to induce immunoglobulin A (IgA) secretion at mucosal cites [31,32], vaccines inoculated via the conventional intramuscular route exclusively induce a systemic immune response characterized mainly by serum immunoglobulin G (IgG) antibodies but barely induce a mucosal immune response. The ability of serum IgG to eliminate virions at the mucosal site might be very limited [33]. Mucosal routes of administration (specifically, nasal or pulmonary delivery) are still considered by the research community to be the ideal and most straightforward approach to inducing potent immune responses against respiratory infections. This is particularly important for prophylactic vaccines to prevent respiratory diseases, such as SARS-CoV-2, RSV, and influenza [34,35,36]. NA-vaccines administered to the mucosal layers, such as the nasal and pulmonary mucosa, could travel to the draining mucosal lymph nodes (LNs) via lymphatic systems under the airway epithelium [37,38,39]. Importantly, the activated mucosal immune system causes the production of pathogen-specific secretory immunoglobulin A (sIgA), which efficiently blocks and neutralizes invading pathogens in mucus at an early stage of infection and acts in the front line of defense against infection [40,41]. Immune cells stimulated by antigens in the respiratory mucosa are also able to produce corresponding specific immune responses to protect against infectious diseases in other mucosal parts of the body (e.g., the gastrointestinal and reproductive tract), which are facilitated by the common mucosal immune system [42]. Moreover, delivery of vaccines to mucosal surfaces can induce systemic immunity with levels similar to those induced by counterparts administered via the conventional parenteral route [43,44]. In addition to expanding immune protection to mucosal sites, mucosal vaccination also brings other advantages. Being an inherently needle-free delivery approach, mucosal vaccine delivery addresses various problems associated with needle injection, such as the risk of needle reuse, which may lead to cross-infections; low patient compliance due to the pain caused by injection; and the necessity of medical staff for the vaccine injection [45].

Although mucosal vaccination is highly appealing, successful delivery of nucleic acids within the airway is extremely challenging because of the complex structures and the harsh microenvironment of the respiratory tract which has evolved to proficiently remove foreign particles (Figure 1B). Compared with that of parenterally administered vaccines, the number of successful mucosal vaccines is very limited. Currently, there is only one nasally-administered vaccine (i.e., Flumist^®^) that has obtained market approval [46], and it is based on live attenuated viruses for the reason that, currently, only virus-based delivery systems (such as attenuated viruses or adenoviruses) have been able to achieve strong respiratory mucosal immune responses [36,47,48]. However, such virus-based vaccines still have many drawbacks, such as safety issues (virulence restoration) and preexisting immunity [46]. Biomaterials-based nonviral delivery systems (Figure 1A), on the other hand, have the advantages of excellent biocompatibility, ease of production, and high flexibility in structure modification and can hence be tailor made for the mucosal delivery of DNA- and IVT-mRNA-based therapeutics [49]. Another advantage of the nonviral delivery platform is its ability to enhance vaccine efficacy by acting as an adjuvant itself or codelivering adjuvants. However, the in vivo delivery efficiency of nonviral materials is usually inferior to that of viruses, impeding the induction of sufficient mucosal immune responses. Successful nasal or pulmonary delivery of NA-vaccines is challenging because of physical and biological barriers at the airway mucosal site. Nevertheless, the field is still developing and evolving, with numerous efforts dedicated to the exploration of various materials for successfully mediating mucosal immunity. Based on our research interest in the airway delivery of genetic payloads, this review highlights recent advances in NA-vaccines inoculated via the nasal or pulmonary routes. We discuss the role of their delivery platforms and summarize crucial factors for the development of successful DNA- or IVT-mRNA-based vaccines that work efficiently in the respiratory mucosa (Figure 1). A successful pDNA or IVT-mRNA delivery system should not only confer protective capacity and efficient mucosal delivery function by overcoming the barriers but improve the translation of antigens from their payload pDNA/mRNA in specific antigen-presenting cells (APCs) or epithelial cells (Figure 1C). This review also covers several relevant topics including the structure of the airway and other barriers to mucosal vaccine delivery and the associated immune system that processes vaccine antigens upon respiratory delivery.

## 2. Barriers of Vaccines Inoculated via Respiratory Route

The respiratory system is critical for mammalian to maintain their life. It carries out the physiological function of exchanging gases between the organism and the environment. Unavoidably, it is subject to the presence of unwelcome foreign substances, such as pollutants and pathogens [50]. One important defense system is the mechanical–physical barrier, constituted by a layer of epithelial cells adhered on the luminal surface of the respiratory tract with tight junctions, preventing the trespassing of inhaled particles [51,52]. Among these epithelial cells, goblet cells secrete a dense, protective mucus that traps foreign particles and pathogens while also disrupting bacterial aggregation (Figure 1). The mucus, carrying trapped foreign substances, is continuously expelled via coordinated movement by the mucociliary “escalator” [53,54]. This mucociliary clearance removes large foreign materials (>5 µm) from the upper respiratory tract (URT). However, pathogens often escape this mechanical–physical barrier, reaching the pulmonary alveoli within the lower respiratory tract (LRT), which may lead to pulmonary inflammation [55]. To prevent such infections, another important line of defense is available. The airway and alveolar fluids contain a variety of soluble substances such as antimicrobial proteins (e.g., lysozymes, lactoferrins, and defensins), microbial opsonins (e.g., complement proteins and surfactant-associated proteins), and enzymes (e.g., proteases and nucleases), building up a robust antiinfection environment within the luminal surface along the entire airway [56].

Because of its important roles in protecting the host from inhaled pathogens, the mechanical–physical barrier poses huge obstacles to vaccines inoculated via the respiratory tract. Since vaccines are also perceived to be exogenous pathogens, intranasally or intratracheally administered vaccines are prone to being expelled via “mucosal clearance”. This, combined with nonspecific binding with positively-charged proteins and enzymatic degradation within the mucus, makes it difficult for NA-vaccines to reach the airway epithelium to activate the mucosal immune system, resulting in limited immune response. To overcome these barriers, a safe and efficient delivery system is imperative for NA-vaccines to be administered via the respiratory tract.

## 3. Mucosal Immune Systems of Respiratory Tract against Infection

### 3.1. The Upper Respiratory Tract

The highly vascularized URT is the primary route of ingress of inhaled pathogens. A dense network of mucosal-associated lymphoid tissues (MALTs) is in the mucosal tissues to help induce pathogen-specific immune responses, reducing occurrences of infections. Peyer’s patches in the small intestine, bronchus-associated lymphoid tissues (BALTs), larynx-associated lymphatic tissues (LALTs), and nasopharynx-associated lymphoid tissues (NALTs) in the rodent nasal cavity are all subsets of MALTs [57,58]. The NALT, commonly regarded to be equivalent to the Waldeyer’s ring in humans [59], also plays a potentially unique role in the initiation of mucosal immune responses based on well-organized local lymphoid structures [60]. Through antigens presented by dendritic cells (DCs) from the parenchyma of nonlymphoid organs and distal sites, the NALT has an important role in generating T helper 1 and T helper 2 cells as well as IgA-secreting B cells in lymph nodes (LNs) [61].

While mucosal DCs capture antigens directly from the luminal side of the airway tract by extending DC dendrites through the tight junctions between the epithelial cells [62], M cells, a group of specialized cells located in the epithelium-overlying follicles of the MALT, are responsible for efficiently transporting antigens from the lumen to the underlying mucosal lymphoid tissues in a process termed “transcytosis” [63]. On their basolateral surface, the membranes of M cells are deeply invaginated, forming pocket structures holding DCs and/or lymphocytes. M cells are perfect antigen-transporting devices, and their reduced lysosome function enables antigen particulates entered from the airway lumen to be effectively delivered to DCs for further processing with minimal modification [63]. Apart from antigen transport, M-cells also aid immune response induction to the transported antigen by releasing a costimulatory signal for T- and B-cell proliferation [64]. Thus, M-cell targeting strategies in the development of mucosal vaccines have grown in popularity as means of improving efficacy in priming mucosal and systemic immune responses.

When captured antigens are transported to draining lymphoid nodes, they are processed into peptides following DC maturation for presentation to antigen-specific CD8^+^ or CD4^+^ cells in the form of major histocompatibility complex (MHC) class Ⅰ or class Ⅱ, respectively. Facilitated by costimulatory molecules and cytokines, the priming of CD8^+^ or CD4^+^ cells initiates programmed cell proliferation and differentiation within draining lymph nodes, accelerating the formation of corresponding antigen-specific effector T cells [65,66]. With the help of cytokines such as IFN-γ, IL-2, and IL-10 secreted by CD4^+^ T helper cells, CD8^+^ effector cells migrate from lymphoid tissues to the site of infection, inducing lysis in infected cells via Fas–FasL interactions or exocytosis of granule-associated proteases containing perforin and granzymes [67,68]. Moreover, a potent and long-term antibody response is built with help from T follicular helper (T_FH_) cells, which provide relevant cytokines and costimulatory molecules for the affinity maturation of B cells, despite the presence of a T cell-independent mechanism for antibody production [69,70,71]. Following isotype switching and the selection of B cells in the germinal center, antigen-specific dimeric immunoglobulin A (dIgA) is generated [72]. Subsequently, transcytosis of dIgA is mediated by polymeric immunoglobulin receptors (pIgRs) expressed on the basolateral surface of the epithelium [73]. Endoproteolytic cleavage of pIgR on the luminal side results in the release of secretory IgA (sIgA) into the lumen [74]. Unlike serum IgA, sIgA is resistant against proteases, thereby being able to protect mucosal surfaces against pathogens despite the protease-rich environment on the nasal mucosa surface [74]. After complete clearance of infectious pathogens, the majority of effector T and B cells undergo apoptosis, leaving behind a small population of memory cells [75,76].

A study on severe acute respiratory syndrome coronavirus (SARS-CoV) showed that memory T cells remain for a long time in patients who have recovered from SARS-CoV infection, conferring a long-term immune protection effect [77]. Memory T cells can be roughly divided into two classes. After clearance of infected pathogens, central memory T cells (T_CM_) typically remain in circulation within lymphoid organs, while effector memory T cells (T_EM_) generally circulate through the red pulp of the spleen and nonlymphoid tissues [78,79]. Both T_CM_ and T_EM_ possess strong capabilities to proliferate and differentiate into effector T cells after encountering a reinfection. T_EM_ in the airway can rapidly remove and act on respiratory pathogens at the early stage of the reinfection [80,81]. However, T_CM_ has to undergo multiple steps, such as activation and differentiation, to initiate significantly delayed effector responses upon encountering pathogens compared to those of T_EM_ [82]. Another population of activated memory T cells does not return to circulation and is therefore known as resident memory T cells (T_RM_). Compared with T_EM_, T_RM_ provides a first line of defense in nonlymphoid tissues against infectious pathogens and limits further spread of infection in the host [83]. Lung T_RM_ seems more effective in promoting a long-term mucosal immune response against airway-transmitted pathogens [84,85]. Respiratory route inoculation has also been suggested to be necessary for generating of T_RM_ cells in the lung [86]. As shown in a previous study, Sendai virus-specific CD4^+^ T_RM_ persists in lung tissues and the airway for several months after infection in C57BL/6 mice, mediating a substantial degree of protection against secondary virus infection [87]. Thus, T_RM_ levels constitute another important indicator of mucosal vaccine efficacy. Given that the mucosal surface is the primary route of ingress of most pathogens, mucosal vaccines providing longer immunity against inhaled pathogens via eliciting robust effector memory populations in mucosal tissues is of great promise.

### 3.2. The Lower Respiratory Tract

Most pathogens that are smaller than 1 µm tend to be able to bypass the mucosal layer on the URT luminal surface, reaching the terminal regions (bronchioles and alveoli) of the LRT. In the lung, the dominant cell population, alveolar macrophages (AMs), accounts for approximately 95% of airspace leukocytes, engulfing most of the antigens [56,88]. Macrophages play an important role in the first line of defense, while CD11c^+^F4/80^−^ DCs seem to be the major APC population in the lung responsible for presenting antigens to pulmonary T cells after infection in mice [89], demonstrating that DCs play an important role in adaptive immune response and in determining the magnitude of pulmonary vaccines’ immune responses.

In alveoli, the recruitment and differentiation of CD11b^+^ DCs can be facilitated by producing type I immune mediators in alveolar epithelial cells (AECs) and immune cells [90]. Pulmonary surfactant (PS)-biomimetic liposomes encapsulating 2′,3′-cyclic guanosine monophosphate-adenosine monophosphate (cGAMP) enter AMs by means of lung specific surfactant protein-A- and -D-mediated endocytosis, after which cGAMP is released into the cytosol and fluxes from AMs into AECs by way of gap junctions [90]. cGAMP, an agonist of the stimulator of interferon genes (STING), stimulates the production of type I interferons in both AECs and AMs, which in turn promotes rapid DC recruitment and differentiation. Through this mechanism, influenza-specific humoral and CD8^+^ T cell immune responses were vigorously augmented in mice after intranasal immunization with PS-cGAMP-adjuvanted H1N1 vaccines [91]. Taken together, these findings suggest that the respiratory delivery of cGAMP as adjuvant might be an alternative strategy for developing pulmonary vaccines.

The antigen-presenting ability of different types of DCs in the respiratory system is also suggested to be different. Although there are relatively larger populations of DCs located in the lung parenchyma and alveolar spaces of the LRT, airway mucosal myeloid DCs are more endocytic and efficient than their aforementioned counterparts for presenting peptide antigens to naive CD4^+^ T cells [92]. Similarly to NALTs, well-developed bronchus-associated lymphoid tissues (BALTs) have been found at branching sites of the bronchial tree in rabbit and feline lungs, while this type of lymphoid structure in human and mice is called inducible bronchus-associated lymphoid tissue (iBALT). The iBALT is induced only by inflammatory stimulation or infection and represents an inducible secondary lymphoid tissue for respiratory immune responses [93]. However, the biological outcomes of the immune response mediated by the iBALT could be beneficial or harmful. In some cases, the persistent exposure of antigens to iBALT during allergic reactions, chronic inflammation, or autoimmune disease may lead to exacerbation of inflammation [94]. Although the local immune mechanism of iBALT is poorly understood, the design of pulmonary vaccines aiming to trigger the formation of iBALT with subsequent protective immunity should be considered. For instance, pulmonary administration of a protein cage nanoparticle promoted the development of iBALT in the lung, resulting in enhanced viral clearance, accelerated induction of viral-specific antibody production, and significantly decreased morbidity and lung damage [95].

In conclusion, nasal and pulmonary administration are the most effective ways to elicit a substantial local immune response in addition to a systemic immune response. Mucosal immunisation stands out for its rapid and comprehensive activation of various immune subsystems in addition to its relatively fewer side effects.

## 4. DNA Vaccines

The first proof of concept for in vivo protein expression with nucleic acids was reported in 1990 by injecting DNA or RNA molecules into mouse skeletal muscle for the expression of chloramphenicol acetyltransferase, luciferase, and galactosidase [96]. It was thereafter demonstrated that the production of cytotoxic T lymphocytes for influenza could be induced by injecting plasmid DNA (pDNA) encoding influenza A nucleoproteins into the quadriceps of BALB/c mice [97]. These pioneer studies confirmed sufficient immunogenicity of DNA vaccines in animal models, providing evidence of this immunization platform’s promising ramifications. DNA vaccines are generally constructed by inserting gene fragments encoding immunogenic antigens into a bacterial plasmid vector, forming pDNA. After pDNA is delivered into the host cell nucleus, antigenic proteins are subsequently expressed. Generally, APCs are the primary targets to be transfected with the genetic material. Following effective presentation in APCs, foreign antigenic proteins initiate specific immune responses [98]. DNA vaccines offer several advantages over conventional vaccines (e.g., live-attenuated, inactivated, or subunit vaccines), as summarized in Table 1. Also, DNA vaccines are generally stable at room temperature (though this may vary between formulations), avoiding the need for an uninterrupted cold chain during storage and transport [99]. Large-scale manufacturing of DNA vaccines primarily involves synthesis of relevant nucleic acids followed by standard cloning into plasmid vectors, avoiding the time- and labor-intensive culturing procedures required by traditional subunit and virus-based vaccines [100]. In contrast to subunit vaccines, DNA vaccines have been demonstrated to induce more potent cytotoxic T cell responses without severe side effects [101]. Although the potential possibility of genome integration remains the primary theoretical safety concern with DNA vaccines, this scenario has not been realized across large numbers of studies and reports [101]. Numerous clinical investigations have also demonstrated DNA vaccines to be largely safe in humans [3,16,102]. All these advantages make DNA vaccines an ideal candidate for rapid responses in the event of epidemic and pandemic outbreaks. Indeed, the first DNA vaccine to be approved for human use was a COVID-19 DNA vaccine (ZyCoV-D) developed in India. It was found to be 67% protective in clinical trials [103], providing evidence that DNA vaccines can be effective in controlling the pandemic [104].

In past decades, DNA vaccines have been studied for the prevention and treatment of a variety of diseases, such as infectious diseases, cancer, autoimmune diseases, and allergies. However, the immunogenicity of DNA vaccines in humans is not as sufficient as that in mouse studies to elicit significant clinical benefits [3]. The poor transport of pDNA into the nucleus of host cells results in low antigen synthesis, limiting the protective immunological responses in the recipient. Several approaches have been explored in recent years to address this issue, including the optimization of codon sequences/transcriptional elements, the incorporation of adjuvants, and enhancing delivery technologies. In this section, we discuss several well-studied delivery vehicles used for DNA vaccines that are administered via nasal or pulmonary routes (Table 2).

### 4.1. Delivery of DNA Vaccines via Respiratory Routes

Intranasal and pulmonary administration of DNA vaccines have attracted widespread attention recently because of various enticing properties. Early reports on inhaled DNA vaccines combined plasmids encoding ovalbumin, hepatitis B surface antigen, and HLA-A*0201-restricted T cell epitopes of *Mycobacterium tuberculosis (M. tuberculosis)*, which resulted in enhanced immunity as indicated by antibodies and cytokine production [105,106]. These pioneering investigations suggested that mucosal immune responses can be more effectively elicited when DNA vaccines are delivered directly to mucosal sites. However, there are still many obstacles that need to be overcome to more effectively utilize mucosal DNA vaccines. Vaccines must penetrate the mucus layer, translocate into target cells, and avoid extracellular and intracellular degradation in order to be effective (Figure 1). For example, delivery via the nasal cavity exposes DNA vaccines to being trapped by the nasal mucus, resulting in enzymatic breakdown. The viscosity and pore size of the mucus layer markedly affect the effective diffusivity of particles on the airway surfaces. Mucociliary clearance from cilia cells also significantly determines the fate of entrapped DNA vaccines. It continuously pushes mucus outwards, expelling mucus from the nasal channel and limiting residence time at the mucosal surface. The dilution effect in bulk mucosal fluids can also impede successful deposition onto the epithelium.

As a result, a safe and effective DNA delivery mechanism must be designed to overcome these obstacles. A suitable delivery system should target mucosal APCs for antigens processing, resulting in selective B and T cell activation. The ultimate goals of DNA delivery systems are to promote uptake of DNA into target tissues and cells, protect DNA from enzymatic breakdown, extend residence time at the target site, boost antigen expression, and optimize immune response, all without sacrificing safety. Section 4.2 discusses in detail several DNA vaccine delivery technologies that have been evaluated for respiratory administration.

### 4.2. Delivery Systems for DNA Vaccines via Respiratory Routes

The most prevalent technologies in the delivery of DNA constructs, such as electroporation, particle bombardment, and jet injectors, have exhibited improved transfection efficiency and huge potential in clinical trials when compared to traditional intramuscular injection of naked pDNA [107,108]. However, these attractive methods are inapplicable for the respiratory route, necessitating the use of a potent delivery strategy to overcome the barriers in the respiratory system. Advancements in nanotechnologies and material science have proven to be beneficial in the effective delivery of pDNA by the synthesis of DNA nanoparticles with diverse structures. Nanoparticles can better pass cell membranes via endocytosis while preventing premature degradation of pDNA and subsequently promote intracellular trafficking into the nucleus following endosomal escape, increasing immunogenicity in animal models and humans. Most importantly, sustained and controlled release of pDNA from nanocarriers at the delivery site recruits APCs, resulting in an improved antigen-specific immune response. The immunogenicity of DNA by vaccines can be further enhanced employing additional adjuvants. Currently, cationic lipids and polymers are two of the most employed nanomaterials for DNA vaccines.

#### 4.2.1. Liposomes and Niosomes

Liposomes and some other vesicular systems are widely used as delivery systems for DNA vaccines. Liposomes generally consist of aqueous cores surrounded by phospholipid bilayers. For antigen delivery, two approved liposomal vaccine formulations used to be available: Inflexal^®^ V (influenza vaccine) and Epaxal^®^ (hepatitis A vaccine). Both of these formulations use virosome-based technology in which viral proteins are bonded to the surface of a liposome carrier similarly to how viral particles are bound [109]. The goal is to imitate a safe viral-like particle capable of eliciting significant protective immune responses. Although both examples have been discontinued, comparable technology could be used to increase the immunogenicity of DNA vaccines. To achieve this, pDNA could be either electrostatically complexed on the surface of cationic liposomes or encapsulated in the aqueous core by a dehydration–rehydration procedure. In general, cationic liposomes have shown higher in vitro transfection efficiency, while nonionic or anionic counterparts have shown enhanced antibody responses in animal models [110,111]. Surface modifications with antigenic components or targeting ligands further enhance immune responses of liposome-based vaccines [111]. It has been shown that liposomes coated with glycol chitosan are mucus adhesive and immune system stimulating [112]. Surface-modified cationic liposomes such as phosphatidylcholine (PC), dioleoyl phosphatidylethanolamine (DOPE), and cholesterol (Chol) elicited stronger humoral, mucosal, and cell-mediated immune responses post intranasal administration in mice against hepatitis than uncoated counterparts [111]. Liposomes containing noncoding pDNA have also been shown to exhibit adjuvant-like behavior, inducing elevated antibody levels and T cell immunity in mice and nonhuman primates [113].

Liposomes have been used as DNA vaccine carriers for intranasal delivery in several investigations to induce efficient immune responses against respiratory pathogens. To induce immune protection against *M. tuberculosis*, D’Souza et al. adopted pDNA encoding antigen 85A formulated with a (+/−)-N-(3-aminopropyl)-N,N-dimethyl-2,3-bis (dodecyloxy)-1-propanaminium bromide (GAP-DLRIE):DOPE liposome [114]. After intranasal immunization in mice, a positive splenic Th1-type cytokine response was induced in GAP-DLRIE:DOPE formulations, but this response was still weaker than that obtained from intramuscular administration with the same dosage. However, the combination of intranasal and intramuscular injections elicited even stronger Th1 type immune responses in the lungs [114]. Another study, performed by Rosada et al., described how a single intranasal immunization with liposome-based formulations of pDNA encoding HSP65 against *M. tuberculosis* led to a remarkable reduction in the amount of bacilli in lungs of mice [115]. The authors employed egg phosphatidylcholine (EPC), DOPE, and 1,2-dioleoyl-3-trimethylammonium-propane (DOTAP) to formulate the delivery system. This formulation also increased the production of IFN-γ and lung parenchyma protection to a level similar to that in mice vaccinated intramuscularly four times the dosage of naked pDNA encoding HSP65 [115]. In addition, intranasal immunization with liposome-based DNA vaccine provided complete protection against influenza after a viral challenge assay [116]. Mice immunized intranasally with liposome-encapsulated pDNA encoding hemagglutinin (HA) protein, but not naked plasmid, were found to produce strong serum IgA/IgG responses and increased IgA titers in bronchoalveolar lavage fluid (BALF) [117]. T cell-proliferative responses were also successfully induced in both intranasal and intramuscular administration [117]. These studies demonstrated the ability of liposomes in the delivery of DNA vaccines inoculated via the intranasal route to confer significant immune protection against respiratory infections in animal models. However, widespread adoption of liposome-based vaccines remains stunted by their relatively lower physical and chemical stability in aqueous dispersions during long-term storage [118]. Accordingly, numerous methods to improve the stability of liposome formulations during storage have been investigated, including freeze-drying, spray-drying, supercritical fluid technology, and lyophilization [119,120,121].

Niosomes, which are nonionic surfactant-based vesicles, have been developed as alternative delivery systems to liposomes because of their advantages such as cost-effective manufacturing, large-scale producibility, and stability [122,123]. Because of their structural similarities to liposomes, niosomes were also applied as vehicles for pDNA, small interference RNAs (siRNAs), and aptamers in target cells [124]. Cationic niosomes, containing cationic lipids, made an effective vector for pDNA delivery and achieved ~95% transfection efficiency in vitro [125]. Later, the same research team reported successful transfection of human tyrosinase gene (pMEL34) and the stability of developed cationic niosomes in transdermal delivery [126]. Perrie et al. reported that niosomes carried with H3N2 influenza virus resulted in enhanced immune response after subcutaneous administration in mice [127]. Mannolysated niosomes encapsulated with pDNA encoding HBsAg were reported to provoke protective immunity against hepatitis B as both a DNA vaccine carrier and adjuvant for oral immunization [128]. However, there have been no reports utilizing niosomes as a mucosal delivery platform in the respiratory tract as far as we know. Their efficacy for the intranasal and pulmonary delivery of DNA vaccine needs further investigation.

#### 4.2.2. Polymers

One of the most appealing characteristics of polymer-based DNA delivery technologies is their flexibility in structure design and modification. Electrostatic interactions allow cationic polymers to form complexes (polyplexes) with DNA vaccines. Polymer synthesis is also relatively inexpensive and simple to scale up. To maximize cellular uptake and transfection effectiveness, the size and surface characteristics of polymeric particles can be adjusted by employing different polymers and methods of preparation [129,130]. It has been found that alveolar macrophages are particularly effective in absorbing particles with diameters ranging from 300 to 600 nm, so the particle size should be less than 3 μm (preferably under 500 nm) for DC-targeted absorption in the respiratory tract [131]. Aside from particle size, particle charge also influences cellular absorption in APCs in the respiratory tract. Where both DCs and macrophages are substantially present, preferential uptake of DNA vaccines into DCs is desirable. DCs can produce large quantities of peptide–MHC II complexes, which are then presented on the cellular surface to initiate T cell activation and differentiation [132]. It has been found that macrophages have higher phagocytic activity than DCs, but also that absorption in DCs can be increased by imparting positive charge to the particle [133]. Polymeric particles can also be modified with functional groups or ligands to improve the cellular uptake of DCs. DCs can preferentially uptake ligand-modified nanoparticles via receptor-mediated endocytosis using C-type lectin receptors or mannose receptors [134].

##### Polyethylenimine

Polyethylenimine (PEI) is one of the most well-studied polymers with high transfection efficiency and has been extensively applied for mediating in vitro and in vivo transfection of DNA molecules [135]. Compared to lipid-based formulations, DNA complexed with PEI has shown improved stability and higher levels of pulmonary transfection, even after nebulization [136,137]. For intranasal immunization, PEI/pDNA complexes encoding SARS-CoV spike proteins induced higher antigen-specific Th2 dominant IgG and IgA antibodies in BALF than naked plasmid counterparts [138]. Cellular immune responses were also detected in a PEI/pDNA treated group, with increased B cells and higher numbers of IFN-γ-, TNF-α-, and IL-2-producing T cells in the lungs [138]. A H5N1 intranasal vaccine with DNA encoding HA formulated with PEI induced potent mucosal and systemic immune responses and elicited both full protection against the parental strain and partial cross-protection against a distinct highly pathogenic strain [139]. Pulmonary immunization of DNA vaccines formulated with PEI also induced robust systemic and CD8^+^ T-cell responses in the gut and vaginal mucosa [140]. Furthermore, mice inoculated via the intratracheal (i.t.) route elicited higher levels of interleukin-2 than those inoculated by intramuscular immunization in lung-associated antigen-specific CD4^+^ T cells [140]. These robust T cell responses, which were induced by i.t. but not intramuscular administration, protected mice from a lethal recombinant vaccinia virus challenge [140]. A similar study also reported that robust pulmonary CD8^+^ T cell populations effectively mediated protective immunity against influenza respiratory challenges after pulmonary immunization with PEI/pDNA [141].

Although PEI appears to be a promising delivery vector for airway inoculated DNA vaccines, one remaining major limitation is its toxicity due to its highly positively-charged and nondegradable nature. Immunization with PEI vaccine was found to provoke the activation of genes with apoptosis, stress responses, and oncogenesis [142]. As a result, biodegradable PEI derivatives with low-toxic profiles have been developed for DNA delivery. A less toxic form of PEI called deacylated PEI (dPEI) with potent transfection efficiency was applied in delivering pDNA encoding HA [143]. Essentially, dPEI is a completely hydrolyzed linear PEI with 11% more free protonatable nitrogen atoms than conventional PEI. Following intranasal administration, dPEI-complexed pDNA vaccine formulations were capable of generating strong systemic and mucosal humoral responses, activating cellular responses, and mediating a higher degree of protection in a challenge study against influenza [143]. Other strategies using covalent modification or electrostatic neutralization of PEI’s cationic group to reduce zeta potential have also been investigated. Poly-lactic-co-glycolic acid (PLGA), a synthetic biodegradable copolymer, has been approved by the Food and Drug Administration (FDA) for human use in delivering therapeutic agents such as proteins and nucleic acids [144]. The negative charge and hydrophobic nature of PLGA could be used to neutralize the positive charge of PEI for safe nucleic acid delivery. Bivas-Benita et al. developed PLGA nanoparticles bearing PEI on their surfaces. Internalization of the DNA-loaded PLGA–PEI nanoparticles was also studied in the human airway submucosal epithelial cell line, Calu-3 [145]. The results suggested that DNA could be detected in the endolysosomal compartment after 6 h incubation with Calu-3 cells and that and the optimal cell viability was achieved when the weight ratio of PEI to DNA was between 1:1 and 0.5:1 [145]. A similar study reported the formulation of PLGA–PEI microparticles, in which 10% PEI (*w*/*w*) efficiently adsorbed DNA and protected DNA from enzymatic degradation [146]. Intramuscular immunization of mice with such PLGA–PEI formulations loaded with pDNA encoding immunodominant antigens of *Listeria monocytogenes* demonstrated that the formulation had an adjuvant effect [146]. These studies indicated that PEI has a favorable profile to be a nonviral gene carrier for DNA vaccines delivered through the respiratory tract, but also that further optimization is still necessary to realize their full potential.

##### Chitosan

Chitosan is a biodegradable and biocompatible polysaccharide derived from chitin and has been frequently employed as a DNA delivery vector because of its biodegradability and biocompatibility [147]. Furthermore, chitosan and its derivatives possess substantial mucoadhesive properties, making them ideal for intranasal administration [148,149]. Chitosan has also been reported to be immune stimulating by enhancing macrophage accumulation and activation, increasing cytokines’ resilience against infections, and promoting cytotoxic T cell response [150,151]. To assess its potential in mediating mucosal immunization, chitosan was used to complex with pDNAs encoding nine different antigens (NS1, NS2, M, SH, F, M2, N, G, and P) from respiratory syncytial virus (RSV) [151]. A single intranasal administration of chitosan–pDNA resulted in a significant reduction of viral titers and viral antigen load in the lungs after an acute RSV infection [151]. In addition, significantly elevated levels of serum RSV-specific IgG antibodies, nasal IgA antibodies, cytotoxic T lymphocytes, and IFN-γ production in the lung and splenocytes were detected in comparison with controls [151]. However, when pDNA encoding the M2 proteins of RSV antigens was formulated with chitosan, virus-specific CTL responses in BALB/c mice were induced only at a level that was comparable to those induced via intradermal immunization [152]. Nonetheless, aerosolized pDNA–chitosan nanoparticles induced higher levels of IFN-γ through pulmonary administration than counterparts immunized by intratracheal and intramuscular administration [106].

In order to achieve targeted delivery of antigen to DCs, biotinylated chitosan nanoparticles loaded with pDNA encoding the nucleocapsid (N) protein of SARS-CoV were developed by Raghuwanshi et al. [153]. Chitosan was modified with bifunctional fusion protein (bfFp) consisting of truncated core-streptavidin fused with anti-DEC-205 single chain antibody (scFv) [153]. The core-streptavid in the arm of bfFp bonded with biotinylated nanoparticles, while anti-DEC-205 scFv imparted targeting specificity to the DCs’ DEC-205 receptors. Intranasal administration of such targeted formulations led to the detection of an enhanced number of N protein-specific systemic IgG and nasal IgA antibodies [153]. In another study, mannosylated chitosan (MCS) formulated with DNA vaccine encoding a multi-T-epitope was employed to facilitate airway delivery and antigen targeting to the APCs in the alveoli [154]. Following intranasal immunization, HSP65-specific sIgA in the BALF was significantly elevated. A modest antigen-specific Th1 (IFN-γ, TNF-α, and IL-2) response and a potent polyfunctional CD4^+^ T response were induced for enhancing mucosal immune protection against *M. tuberculosis* in the spleen and lung, respectively [154].

Thiolated chitosan derivatives have been found to improve the transfection efficiency of chitosan for intranasal delivery. In a study performed by Bernkop-Schnürch et al., thiol-bearing moieties were introduced on the polymeric backbone of chitosan in order to prepare thiolated chitosan that could interact with mucus glycoproteins via the formation of disulfide bonds [155]. The results indicated that thiolated chitosan improved mucosa adhesiveness to the mucus layer 6- to 100-fold compared to the unmodified counterpart, resulting in enhanced mucus permeation. Simultaneously, increased penetration at the mucosal surface was also observed in chitosan-coated PLGA nanoparticles encapsulating macromolecules [156,157]. Based on this fact, an emulsion–diffusion–evaporation technique was employed to prepare cationic nanospheres composed of biodegradable and biocompatible copolyester PLGA, with a PVA–chitosan blend stabilizing the PLGA nanospheres [158]. Despite the charge on the nanospheres being sufficient to bind the negatively charged DNA, the immunity of DNA vaccines complexed by this formulation remains to be illuminated. In one study, chitosan-coated PLGA was employed to deliver pDNA encoding foot-and-mouth disease (FMDV) capsid protein and bovine IL-6 to protect mice against FMDV infections [159]. This chitosan/PLGA/pDNA vaccine formulation provided enhanced protective immunity against FMDV post-intranasal immunization [159].

In a recent publication, chitosan was adopted to coat star-shaped gold-nanoparticles to yield a gold-nanostar chitosan (AuNS–chitosan) nanoformulation for intranasal delivery of a DNA vector expressing S protein of SARS-CoV-2 [160]. Six-time repeated dosing of AuNS–chitosan DNA vaccine induced high levels of S protein-specific IgG, IgM, and IgA antibodies in serum up to 8 weeks in both BALB/c and C57BL/6 mice as assessed using an ELISA assay [160]. IgG and IgA sustained their levels until week 8, then rose back to a high level with a single 7th dose in week 14. The serum neutralization IC_50_ (serum dilution factor) for infectivity inhibition concentrations were determined to be 1:83.8, 1:47.5, and 1:150 (collected in week 18 of the study) for pseudoviruses engineered with S proteins from SARS-CoV-2-Wuhan, SARS-CoV-2-beta mutant, and SARS-CoV-2-D614G mutant variants, respectively [160]. With the help of immunophenotyping and histological analyses, the authors further revealed chronological events involved in the recognition of S antigen by resident DCs and alveolar macrophages in the lungs of DNA vaccine transfected mice [160]. These APCs further primed the draining lymph nodes and spleen for peak antigen-specific cellular and humoral immune responses. Although this proof-of-concept study suggested the capabilities of the AuNS–chitosan DNA vaccine to elicit potent mucosal immune responses, further development with a reduced dosing regimen and more key evidence (e.g., sIgA levels in BALF samples, details in the subsets of effector T cells and memory T cells, the protection efficiency of SARS-CoV-2 challenge studies, etc.) will be required to validate its potential for clinical translation.

**Table 2 nanomaterials-12-00226-t002:** Summary of DNA vaccines inoculated via respiratory tract.

Disease	Nanoparticle	Coding Antigens	Experimental Animal	Administration	Immune Response ^1^	Ref.
Hepatitis	PC/DOPE/Chol	S protein	mice	i.n.	HIR(+)/MIR(+++)/CIR(+)	[111]
Tuberculosis	GAP-DLRIE:DOPE	85A	mice	i.n.	Th1 CIR(+)	[114]
Tuberculosis	EPC/DOPE/DOTAP	HSP65	mice	i.n.	Th1 CIR(++++)	[115]
Tuberculosis	MCS	HSP65	mice	i.n.	MIR(+++)/CIR(++)	[154]
Tuberculosis	Chitosan	Multiantigens	HLA-A2	i.t.	CIR(++)	[106]
Influenza	DODAC/DOPE/PEG	HA	mice	i.n.	HIR(++)/MIR(+)	[116,117]
Influenza	PEI	HA	mice	i.n.	HIR(+++)/MIR(++)	[139]
Influenza	dPEI	HA	mice	i.n.	HIR(++++)/MIR(++++)/CIR(+)	[143]
SARS-CoV	PEI	S protein	mice	i.n.	HIR(+++)/MIR(+++)/CIR(++)	[138]
SARS-CoV	Chitosan	N	mice	i.n.	HIR(++++)/MIR(++++)	[153]
HIV	PEI	HXBc2 gp120	mice	i.t.	CIR(++)	[140,141]
RSV	Chitosan	Multiantigens	mice	i.n.	HIR(++++)/MIR(++++)/CIR(+)	[151]
RSV	Chitosan	M2	mice	i.n.	CIR(+)	[152]
COVID-19	Chitosan–gold	S-protein	Mice	i.n.	MIR(N.A.)/HIR(++)/CIR(+)	[160]

^1^ Responses are geometric means of postvaccination increases in specific antibodies versus control in vaccine recipients: ++++, >10-fold; +++, 5- to 10-fold; ++, 2.5- to 5-fold; +, 1.5- to 2.5-fold. RSV: respiratory syncytial virus; HA: hemagglutinin protein; HIV: human immunodeficiency virus; HIR: humoral immune responses; MIR: mucosal immune responses; CIR: cellular immune responses; SARS-CoV: the severe acute respiratory syndrome coronavirus; i.n.: intranasal administration; i.t.: intrathecal administration; N.A.: not available.

## 5. IVT-mRNA Vaccines

In 2019, the outbreak of COVID-19 caused by SARS-CoV-2 spread throughout the world, developing into a global pandemic. Application of safe and effective vaccines is expected to be the most efficient medical approach in controlling and stopping the pandemic of COVID-19. BNT162b2 (BioNTech/Pfizer, Mainz, Germany/New York, NY, USA) and mRNA-1273 (Moderna, Cambridge, MA, USA) are both lipid nanoparticle-formulated, nucleoside-modified IVT-mRNA vaccines that received approval for emergency use by the FDA with extremely high protection rates against COVID-19 [25,26,27,161]. The area of IVT-mRNA vaccines is rapidly evolving; a considerable amount of clinical evidence has been gathered over the last several years, widely establishing IVT-mRNA vaccines as a highly promising medical strategy [162]. The design and synthesis of IVT-mRNA and associated delivery technologies are key to the success of IVT-mRNA vaccines (Figure 2). IVT-mRNA encoding specific proteins of interest (POI) is transcribed in vitro according to a linearized DNA template. Once delivered into host cells, IVT-mRNA utilizes the translation machinery of the host to produce corresponding POI in cytoplasm without the need to enter the nucleus to be functional [5]. If the POI is an appropriate antigen, the resulting POI induces antigen-specific immune responses following effective antigen presentation by APCs. Currently, two major classes of IVT-mRNA have been investigated broadly as vaccines: conventional nonreplicating mRNA and self-amplifying mRNA (saRNA). Both share elements of a eukaryotic mRNA that are essential for translation and stability: i.e., a cap structure (m7Gp3N), 5′- and 3′- untranslated regions (UTRs), an open reading frame (ORF), and a poly(A) tail. Compared with conventional nonreplicating mRNA, saRNA has the potential to produce more antigen protein with the viral replication machinery. The design and development of saRNA vaccines with validated immunogenicity and efficacy has been recently reviewed elsewhere [163]. For IVT-mRNA-based therapeutics, robust delivery systems are generally necessary for the successful transport and uptake of IVT-mRNA into target organs. The instability, innate immunogenicity, and inefficient in vivo delivery to organs beyond the liver and spleen are major problems that need to be solved in the future applications of IVT-mRNA. As a result, a wide range of in vitro and in vivo delivery systems have been designed. Some vaccine formulations contain adjuvants proven to potentiate subunit vaccines [164,165,166], while others generate strong responses despite the absence of known adjuvants. Furthermore, highly effective RNA carriers with low toxicity have been produced, allowing for sustained antigen expression in vivo in some instances [11,12,167]. Apart from that, rational design of RNA sequences has also greatly improved the potential of IVT-mRNA. In this section, we briefly summarize the recent advances in conventional IVT-mRNA vaccines regarding their structure optimization and specifically focus on the delivery system of IVT-mRNA vaccines inoculated via the respiratory tract route (Table 3).

### 5.1. Structural Optimization of IVT-mRNA

Because of mRNA’s susceptibility to nuclease degradation and its high innate immunogenicity, optimization of IVT-mRNA structural elements is crucial to enhance the expression of POI. Over the past decades, a great deal of effort has been given to modifying the structural elements of IVT-mRNA in order to improve its stability and translation efficiency. As extensively discussed in previous reviews [1,5], a functional 5′-cap structure required for the initiation of translation can be added in vitro via either post- or cotranscriptional processes. Vaccinia virus capping enzyme (VCE) is the most widely used commercially available reagent for posttranscriptional addition. First, IVT-mRNA can be modified with a Cap 0 (m7GpppN) structure using VCE. Then, using 2′ *O*-methyltransferase, Cap 0 can be switched to Cap 1 (m7GpppmN), which enhances the translation efficiency of IVT-mRNA [161,162]. However, large-scale manufacturing of IVT-mRNA may consume a huge amount of enzymes using this approach, which would increase the production costs and limit the approach’s widespread application [5]. Alternatively, the 5′-cap can be added simultaneously during the in vitro transcription process of IVT-mRNA with cap analogs (e.g., m7GpppG). However, this method can lead to inefficient capping and reversed cap orientation (i.e., Gpppm7G) [163,168]. To overcome this problem, anti-reverse cap analogues (ARCAs; m27,3′-OGpppG) can be introduced into the in vitro transcription and enhance IVT-mRNA translation efficacy [169,170]. For cotranscriptional mRNA capping, an ARCAs to GTP ratio of 4:1 in a single reaction is optimal for maximizing both RNA yield and the proportion of capped transcripts with commercial transcription kits, yielding up to 100% capping efficiency [171]. Recently, CleanCap^®^, a method developed by TriLink Biotechnology, was reported to generate IVT-mRNA with high translational efficiency, simplifying the production of 5′-capped IVT-mRNA. Based on these advantages, the approach of CleanCap^®^ has been used for 5′ capping of BNT162b2 and mRNA-1273 vaccines [161,172].

In synergy with a 5′-cap, the poly(A) tail regulates translation efficiency and stability of IVT-mRNA. Incorporation of poly(A) can be introduced by recombinant polymerase or by direct transcription from a DNA template with a defined length of poly(T) nucleotides [173]. Although the incorporation of modified nucleotides into the poly(A) tail can be introduced with recombinant poly(A) polymerase to inhibit deadenylation mediated by poly(A)-specific nucleases, a mixture of mRNAs with inconsistent lengths of RNA poly(A) tail is the limitation for this approach, especially for clinical applications [174]. Therefore, the optimal length of the poly(A) tail, around 100–150 nucleotides, is preferably added from the encoding DNA template [175]. However, poly(T) nucleotides in the plasmid template are often lost during the passaging of *E. coli*. Thus, adding a precise length of poly(A) to IVT-mRNA is still a challenge for IVT-mRNA production.

Optimization of the 5′- and 3′- UTRs is also important to enhance the expression and stability of IVT-mRNA [176]. UTRs contain regulatory sequence elements usually harbored by viral or eukaryotic genes, which can be designed via de novo methods or deep-learning methods [5,177]. Previous studies have shown that optimization on AU-rich elements and G–C ratios increase the stability of IVT-mRNA and duration of protein expression [178,179]. Furthermore, Steve Pascolo et al. added an eIF4G-recruiting aptamer sequence to the 5′ UTR of IVT-mRNA in order to increase the expression of IVT-mRNA [180]. Recently, a sequence of IVT-mRNAs was systematically engineered through deep-learning methods, significantly enhancing the expression of SARS-CoV-2 antigens [177]. However, codon optimization replacing rare codons with frequently used synonymous codons may not be necessary. This approach will affect the generation of potent cryptic T cell epitopes for some IVT-mRNA encoded vaccines [181,182], thus affecting the proper folding of some proteins that need a slow translation process because of rare codons [183].

IVT-mRNA is inherently immunogenic, it activates pattern recognition receptors (PRRs) such as Toll-like receptors (TLR-3, TLR-7, and TLR-8) in the endosomal compartments of host cells [184,185]. In some cases, activation of innate immunity by IVT-mRNA is potentially advantageous for vaccines, as it promotes recruitment of DCs to the administration site and subsequent maturation of DCs to elicit potent adaptive immunity. However, this feature of IVT-mRNA may also lead to improper or excessive activation, resulting in the inhibition of antigen expression and negative immune responses [186]. Currently, modified nucleotides (e.g., pseudouridine and 5-methylcytidine) can be incorporated into IVT-mRNA during transcription, and chromatographic purification methods have been used to remove contaminating double-stranded RNA after transcription [187,188,189]. This approach has been shown to effectively avoid undesired immune stimulation, subsequently increasing production of POI [190]. This nucleotide modification technology has been successfully employed in the production of the BNT162b2 (BioNTech and Pfizer) and mRNA-1273 (Moderna) COVID-19 mRNA vaccines.

### 5.2. Delivery Systems for IVT-mRNA Vaccine Inoculated via the Respiratory Route

In addition to structural optimization and high-quality manufacturing of IVT-mRNA, another major technical barrier in the clinical translation process of IVT-mRNA therapeutics is the development of safe and efficient delivery systems. Whether a potent protective immunity can be established by an IVT-mRNA vaccine is largely attributed to the choice of a safe and efficient delivery system. Basically, a safe IVT-mRNA delivery system should at least condense IVT-mRNA, so that the fragile molecule can be protected from degradation in extracellular space, and facilitate endosomal escape following cellular uptake in cytosol. The successful delivery of IVT-mRNA to the lungs at low dosages would reveal a meaningful region for therapeutics or vaccines designed for pulmonary application. This endeavor, however, remains difficult, because nanocarriers that carry IVT-mRNA to the lungs behave differently than counterparts using the parenteral route. For example, the potency of IVT-mRNA may be significantly compromised by the nebulization process. Despite the fact that modern nebulizers are designed to gently aerosolize medicine, nanoparticles are nevertheless subjected to shear force that could compromise the structures of vehicle–mRNA complexes. Biological aspects of the airway pose other barriers against nebulized IVT-mRNA administration. The cells, proteins, biomolecules, and physical barriers with which nanoparticles interact when delivered via nebulization differ from those with which they interact in bloodstream [191,192,193]. Airway cells are also highly heterogeneous, which may further influence delivery properties [194]. Nevertheless, there have already been several reports on the successful delivery of IVT-mRNA to the airways.

#### 5.2.1. Lipids

Currently, the most successful and frequently used delivery vehicle for the in vivo application of IVT-mRNA is lipid nanoparticles (LNPs) which consist of distinct components with variable proportions [195]. For most LNPs, the ionizable lipid is a key element responsible for packaging the negatively charged IVT-mRNA and enabling the endosomal escape of IVT-mRNA molecules into the cytoplasm. A recent publication from Norbert Pardi et al. suggested that LNP formulations have favorable immunostimulatory profiles that promote the induction of strong responses from T_FH_ cells, germinal center B cells, long-lived plasma cells, and memory B cells that are associated with durable and protective antibodies in mice. The authors further revealed that the adjuvant activity of the LNP relied on the ionizable lipid component and IL-6 cytokine induction [196]. The resulting magnitude and duration of IVT-mRNA expression were favorable for inducing potent immune responses, as evidenced by a study performed by Tam et al. The authors found that exponentially increasing dosing profiles led to prolonged antigen retention in lymph nodes and increased numbers of T_FH_ cells and germinal center B-cells, as well as eliciting >10-fold increases in antibody production relative to single bolus vaccination postprime [197]. Apart from ionizable lipids, phospholipids play a structural role in the formation and disruption of the lipid bilayer. Neutral lipids, such as cholesterol, are usually used as a stabilizing element for increasing transfection efficiency. Lipid-anchored polyethylene glycol (PEG) is a biocompatible and inert polymer, sterically stabilizing the LNPs and reducing nonspecific binding to proteins for increased circulation time in vivo. LNPs were originally developed as highly efficient carriers of short-interfering RNAs (siRNA) to hepatocytes via intravenous administration [198]. Intramuscular, intradermal, and intratracheal administration of LNP-encapsulated mRNA have been found to produce higher and more prolonged protein levels at the site of inoculation than those produced by systemic delivery [195]. However, LNP needs to be specifically designed and optimized for the efficient pulmonary delivery of IVT-mRNA [199]. In one study, cationic LNPs and mannose-conjugated LNP (Man-LNP) were separately applied to encapsulate IVT-mRNA encoding the HA gene of the H1N1 influenza A virus. Following intranasal administration, both formulations induced HA-specific responses with high levels of IgG2a subtype and enhanced the production of both Th1-associated IFN-γ and Th2-associated IL-4 [34]. However, Man-LNP induced significantly higher hemagglutinin inhibition titers than cationic LNP-based counterparts did after boosting immunization [34]. Furthermore, LNP represents a powerful technology for the respiratory delivery of therapeutic genes. LNP-carrying chemically modified IVT-mRNA encoding cystic fibrosis transmembrane conductance regulator (CFTR) was successfully applied for the treatment of cystic fibrosis following nasal administration in CFTR knockout mice. The chloride response in two consecutive doses of CFTR-mRNA/LNP-treated mice recovered up to 55% of the levels observed in wild-type mice and lasted up to 2 weeks posttransfection [200]. Motivated by the huge unmet needs for lung IVT-mRNA delivery vehicles, as well as by the lack of established LNP design principles, Lokugamage et al. recently reported an in vivo cluster-based iterative screening approach to identify LNP chemical traits that promote IVT-mRNA lung delivery [199]. They discovered that a low PEG molar ratio increased the performance of LNPs with neutral helper lipids, whereas a high PEG molar ratio improved the performance of cationic helper lipids, uncovering and optimizing LNPs for low-dose IVT-mRNA delivery.

The optimized LNP nebulized delivery of an IVT-mRNA generating a broadly neutralizing antibody against hemagglutinin protected mice against a fatal challenge of the H1N1 subtype of the influenza A virus and delivered IVT-mRNA more effectively than LNPs previously tuned for systemic delivery. Several limitations of this study should be kept in mind: (1) the findings focused on LNPs containing 7C1, but more research is needed to quantify the link between LNP composition and nebulized delivery; (2) more studies are needed to quantify the link between payload and delivery efficiency, as the protein expression kinetics observed may vary depending on the IVT-mRNA payload; (3) these findings, from mice models, may or may not translate predictably to larger animals, such as nonhuman primates; (4) LNP size and zeta potential, as well as other altered physiologies associated with diseased pulmonary tissues, may impede pulmonary LNP delivery.

#### 5.2.2. PEI

PEI has also been proven to be functional as a potent mucosal adjuvant for intranasal administration [201]. However, how to maximize its adjuvant effect and how to minimize its toxicity remain open challenges. Optimizing the chemical structure of PEI seems to be a practical solution to this problem. Cationic cyclodextrin–PEI conjugate was developed by Li et al. and complexed with anionic IVT-mRNA encoding ovalbumin (OVA) through electrostatic interactions [39]. This formulation showed the ability to stimulate DC maturation and migration after intranasal inoculation in mice, further enhancing both humoral and cellular immune responses [39]. In addition, a cyclodextrin–PEI-based formulation was further evaluated for its ability to penetrate the airway epithelial barrier and its paracellular delivery efficiency using IVT-mRNA encoding HIV gp120 [202]. With prolonged residence in the nasal cavity and excellent intracellular delivery, potent systemic and mucosal anti-HIV immune responses were induced after intranasal dosing [202].

#### 5.2.3. Other Nonviral Vectors

Chitosan was reported to condense saRNA encoding hemagglutinin and nucleoprotein of influenza virus, and it expressed antigen in DCs after SC injection [203]. Recently, intranasal delivery of chitosan nanoparticles encapsulating mRNA encoding influenza HA2 and M2e antigens was reported to elicit efficient protective immune responses against avian influenza in chickens [167]. The aforementioned AuNS–chitosan nanoformulation also showed robust delivery of IVT-mRNA encoding firefly luciferase (fLuc-mRNA) in the lungs of mice upon intranasal delivery as measured by bioluminescence imaging [160]. Using a similar fLuc system, Yingshan Qiu et al. reported that PEG_12_KL4 (a monodisperse linear PEG of 12-mers attached to synthetic cationic KL4 peptide) formed nanosized complexes with fLuc-mRNA and that the intratracheal administration of PEG_12_KL4/fLuc-mRNA complexes resulted in luciferase expression in the deep lung region of mice 24 h posttransfection that was superior to expression induced by naked IVT-mRNA and Lipofectamine 2000 based lipoplexes [204]. However, both of these studies using luciferase reporter systems did not evaluate the antigen-specific immune responses induced by the IVT-mRNA formulations [160,204], making it difficult to affirm whether these delivery systems are suitable for the application of IVT-mRNA vaccines.

Meanwhile, lipid/polymer hybrid complexes have also been employed for efficient pulmonary delivery of IVT-mRNA. Compared with single lipid or polymer delivery systems, a stable IVT-mRNA vaccine particle could be formulated using positively charged protamine to concentrate IVT-mRNA followed by encapsulation with DOTAP/Chol/DSPE-PEG [205]. Such liposome/protamine hybrid complexes exhibit stronger capacities to stimulate DCs’ maturation, which further induces potent antitumor cellular immune responses after intranasal administration with IVT-mRNA encoding cytokeratin [205].

Some commercial transfection reagents have also been employed for IVT-mRNA vaccine delivery via the respiratory tract. Intranasal administration of mRNA encoding HSP65 protein dissolved in Ringer’s lactate solution prompted the specific production of IL-10 and TNF-α in the spleens of mice, and protected the mice from subsequent challenge with *M. tuberculosis* [206]. In both prophylactic and therapeutic immunization models, IVT-mRNA delivered in nanoparticles prepared with Stemfect^TM^ induced tumor immunity correlated with splenic antigen-specific CD8^+^ T cells, while naked counterparts failed to elicit antigen-specific immune responses [207].

**Table 3 nanomaterials-12-00226-t003:** Summary of RNA vaccines inoculated via the respiratory tract.

Disease	Nanoparticle	Coding Antigens	Model Tested	Administration	Immune Response ^1^	Ref.
Influenza	LNP	HA	mice	i.n.	HIR(+)/CIR(++)	[34]
Influenza	Chitosan	HA and M2	chicken	i.n.	HIR(++)/MIR(++)/CIR(+)	[167]
HIV	Cyclodextrin–PEI conjugate	gp120	mice	i.n.	HIR(+)/CIR(+)	[202]
Model antigen	Cyclodextrin–PEI conjugate	OVA	mice	i.n.	HIR(+)/MIR(+)CIR(+)	[39]
Aggressive Lewis lung cancer model	Cationic liposome/protamine	cytokeratin 19	mice	i.n.	CIR(+)	[205]
Tuberculosis	Ringer’s lactate solution	HSP65	mice	i.n.	CIR(++)	[206]
E.G7-OVA tumor	Stemfect mRNA transfection reagent	OVA	mice	i.n.	CIR(++++)	[207]

^1^ Responses are geometric means of postvaccination increase in specific antibodies versus control in vaccine recipients: ++++, >10-fold; +++, 5- to 10-fold; ++, 2.5- to 5-fold; +, 1.5- to 2.5-fold. HA: hemagglutinin protein; M2: matrix protein 2; OVA: ovalbumin; HIV: human immunodeficiency virus; HIR: humoral immune responses; MIR: mucosal immune response; CIR: cellular immune responses; SARS-CoV: severe acute respiratory syndrome coronavirus; i.n.: intranasal administration.

## 6. Conclusions and Perspective

The mucosal immune system provides a first line of defense in the host’s resistance to pathogen invasion and controls further infections at peripheral tissues. For the delivery of NA-vaccines, mucosal routes could be as good as or even better than parenteral counterparts. However, the complexity of the pulmonary environment poses huge obstacles for vaccines in inducing mucosal immunity compared with the parenteral route. This is especially difficult for NA-vaccines when considering their distinct characteristics, such as their high susceptibility to degradation by nuclease, negative charge, and high molecular weight. To overcome the mechanical–physical barriers of the respiratory tract, developing safe and efficient delivery systems is crucial for the successful clinical translation of NA-vaccines, as proven by the LNP delivery system in COVID-19 mRNA vaccines. Because of their similarity to microorganisms in both size and structure, nanodelivery vehicles play multifunctional roles in NA-vaccines by acting as a cargo carriers and potentially also as adjuvants [196]. Such nanosized systems can augment mucosal immune responses via simulating the natural infection process. For DNA vaccines, efficient delivery systems can improve immune responses by enhancing pDNA delivery into the nuclei of the host cells, which increases the expression of antigens. For mRNA vaccines, efficient carriers protect mRNA from premature degradation while optimizing in vivo expression of antigens, inducing a potent protective immunity. Although evidence has shown that NA-vaccines inoculated via the respiratory tract could elicit strong and long-term mucosal, humoral, and cell-mediated immune responses in animal models, there is still no NA-vaccine approved for mucosal vaccination purposes. As a result, innovative mucosal delivery technologies that could mediate successful clinical translations are urgently needed. Based on previous studies and our experiences, an ideal delivery system could provide the NA-vaccine with further unique features: (1) an electrically neutral structure with a hydrophilic shell, to bestow an efficient mucus-penetrating ability and to mediate rapid deposition onto epithelial surfaces; (2) APCs or M-cell targeting, to enhance the uptake of nucleic acids; (3) structures with protonated ability and/or membrane disruptive properties, which may facilitate efficient endosomal escape, avoiding enzymatic degradation.

Another interesting but scarcely discussed topic is the determination of the more advantageous route (between intranasal or intratracheal inoculation) for NA-vaccines. As shown in Table 2 and Table 3, most studies have chosen the nasal route because of its convenient and noninvasive features. Nevertheless, there are still certain concerns regarding nasal administration. For example, negative perception for nasal vaccines was generated from reported cases of Bell’s palsy after intranasal dosing of inactivated influenza vaccines [208,209]. Thus, neurotoxicity tests are necessary for nasal NA-vaccines in order to confirm their safety profiles. On the other hand, the intratracheal route also suffers from limitations. For example, it is not possible to apply intratracheal instillation in humans because of poor compliance and its invasive characteristics, the latter of which may cause unnecessary damage to the airway. Alternatively, one of the most appropriate approaches of delivering NA-vaccines to human airway would be nebulized formulations. Not only do nebulized formulations tend to be more evenly distributed throughout the respiratory tract, but the inhalation of aerosolized vaccine is also highly acceptable and tolerable for the recipient [210]. Unfortunately, nebulization of nucleic acid-based formulations tends to be inefficient [211]. A previous study found that as little as 10% of the nucleic acid payload in a nebulization device chamber could be successfully emitted [212]. Advanced nebulization strategies and optimized formulations have significantly improved the situation; now, even fragile IVT-mRNA has been successfully utilized in aerosolized formulations for in vitro and in vivo investigations [213,214]. As a result, nebulized formulations appear to be a feasible and attractive approach for the pulmonary delivery of NA-vaccines.

In summary, the field of respiratory mucosal vaccine has been bolstered by advances in novel nucleic acid formulations based on nanotechnologies. Thus, we hope that successful clinical nasal or pulmonary administration of NA-vaccines will be on the horizon, revolutionizing the vaccine development field as IVT-mRNA vaccines did during the COVID-19 pandemic.

## Figures and Tables

**Figure 1 nanomaterials-12-00226-f001:**
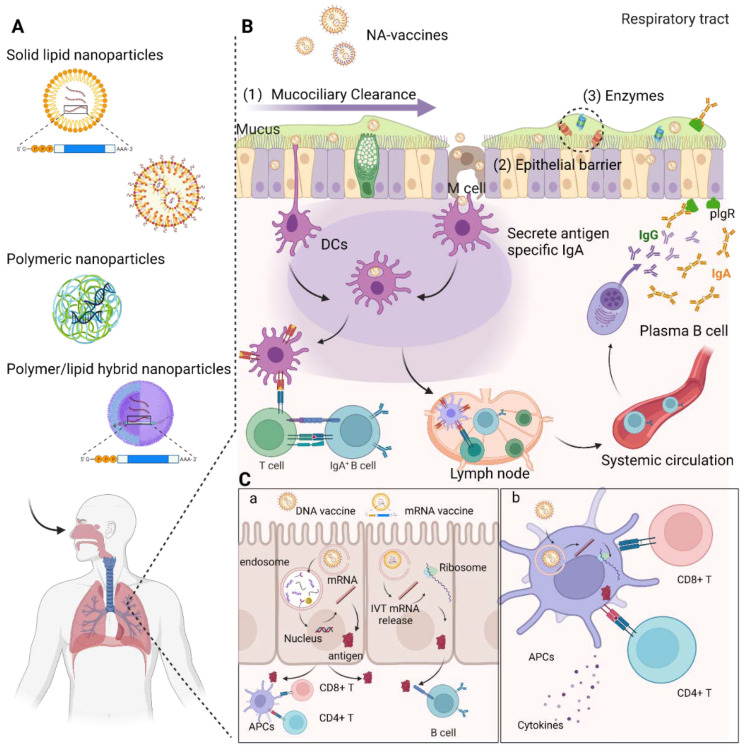
Overview of nucleic acid-based (NA-) vaccines administrated via the respiratory tract using nanotechnologies. (**A**) Schematic view of different nanoparticles used for intranasal and pulmonary vaccinations. (**B**) Physical and biological barriers at the airway mucosal site and mechanism of immune responses in the respiratory tract mediated by mucosal-associated lymphoid tissues (MALTs). NA-vaccines transcytose from the mucus layer into the epithelial tissues by microfold cells (M cells) or passively diffuse through epithelial cell junctions. Other NA-vaccines are captured and internalized by APCs, such as DCs, from their extension through epithelial junctions. APCs that have been transfected with genetic antigens migrate to the nearest lymph node to activate T cells and B cells. Activated B cells proliferate in the lymph node and enter the systemic circulation to the mucosal effector sites. B cells locally differentiate into antibody-secreting plasma cells to produce IgA dimers. IgA dimers are secreted via pIgR at the mucosal surface. Antigen-specific systemic IgG is also produced. (**C**) NA-vaccines are taken up by epithelial cells (**a**), and pathogen-derived antigens are then transcribed and translated from plasmid DNA or IVT mRNA and secreted into the extracellular space, where they can be taken up by professional APCs such as DCs. (**b**). APCs then present antigens to naïve T cells for activation and differentiation, promoting humoral and cell-mediated immune responses against the encoded antigen.

**Figure 2 nanomaterials-12-00226-f002:**
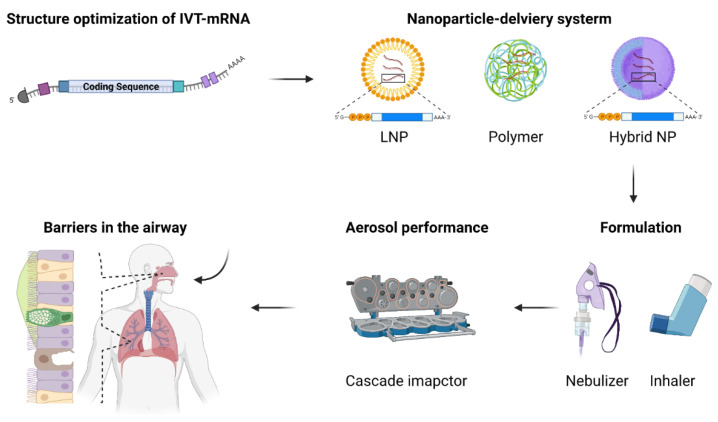
Schematic illustration of key factors associated with the technology development of IVT-mRNA vaccines via the respiratory route. IVT-mRNA structural elements, including elongation of the poly(A) tail, the 5′ cap, the structure of UTRs, and the ORF with optional incorporation of modified nucleotides, are optimized to enhance the stability and reduce the innate immunogenicity. Nanoparticle-based carriers are designed to facilitate the delivery of IVT-mRNA across the barriers in the airway. Liquid aerosol or dry powderformulations are then developed with the identification of a suitable inhalation device (nebulizer or powder inhaler) for clinical applications. The aerosol performance, IVT-mRNA stability after aerosolization, immunogenicity, and vaccine efficacy of the inhaled formulation should be thoroughly characterized and evaluated.

**Table 1 nanomaterials-12-00226-t001:** Advantages of NA-vaccines compared with conventional vaccines.

Category	DNA Vaccines	RNA Vaccines
Design	Rapid design with the coding sequence of antigens A single formulation with multiple antigens is possible	Rapid design with the coding sequence of antigens A single formulation with multiple antigens is possible
Production	Rapid and reproducible production based on in vitro bacterial culture Large-scale manufacture without inactivation of infectious pathogens or purification of recombinant antigens Antigens with proper folding are produced in vivo	Rapid and reproducible production based on in vitro transcription Large-scale manufacture with “cell free” process Antigens with proper folding are produced in vivo, with the cytosol as its target, only transiently expressed
Stability	Depends on the formulation Ease of storage and transportation in most cases	Depends on the formulation Cold chain transportation is generally required
Immune responses	Both cellular and humoral immune responses	Both cellular and humoral immune responses without risk of genome integration

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
