# Peer review of "Nanotechnologies in Delivery of DNA and mRNA Vaccines to the Nasal and Pulmonary Mucosa"

_nanomaterials, 2022, doi:10.3390/nano12020226_

Round 1

Reviewer 1 Report

Sorry, I read only the first three pages, as there are too many imprecisions and confusions already at this stage. Thus, I would suggest that the authors check again their review, eventually involve more experts to help them, revise the manuscript and re-submit.

Here are my suggestions on the first 3 pages:

Introduction. The first lines tell about DNA and mRNA vaccines equally, which nowadays does not seem appropriate since clearly the mRNA vaccines have shown their great potential while plasmid DNA vaccines did not.

“Since host cells are responsible for antigen production, correct folding of the protein and natural glycosylation are required”, you mean “..are guaranteed”?

What is “the target site within host cells”?

Delivery vehicles or device are imperative. It is not true, a lot of pre-clinical and clinical studies have shown vaccine efficacy of naked plasmid and mRNA vaccines. Delivery vehicle and device improve those efficacies

Nucleic acid-based vaccines use (not uses)

Can scale up rapidly

Any disease with effective and well characterized antigens. It is not the disease but the pathogen that has antigens

Expansive?

The mRNA vaccines did not display extremely high protection rates against SARS-CoV-2 infection and were not initially tested for this. They display extremely high protection rates against COVID-19.

It is not 94-95 % protection against SARS-CoV-2 infection but 94-95% protection against COVID-19

Table 1: mRNA vaccines are not at all “extremely” unstable: The thawed, undiluted COVID-19 vaccine can be stored at fridge temperatures of 2 °C to 8 °C for one month (31 days), https://investors.biontech.de/news-releases/news-release-details/update-stability-comirnatyr-longer-storage-possible/. In addition, it is the liposome not the mRNA that is unstable (pure mRNA is very stable even at room temperature). Thus, if for example a plasmid DNA would be formulated in a liposome, such a vaccine will also be unstable. To conclude mRNA and plasmid DNA are both stable molecules but the formulations of those nucleic acids may use (as it is the case for the current mRNA vaccines) unstable particles.

Reviewer 2 Report

The manuscript "Nanotechnologies in Delivery of DNA and mRNA Vaccines to the Nasal and Pulmonary Mucosa" is a review on the use of nanotechnologies for the delivery of delicate molecules for added-value medical applications. The topic is particularly interesting both for the scientific community and for society. Moreover, this review is well organized and an accurate study of the literature has been performed. Therefore, the publication is recommended; but after some revisions, as follows:

  • Abstract. Add more information on the kind of nanotechnologies required for the loading and delivery of DNA and mRNA vaccines to the nasal and pulmonary mucosa.
  • Add an abbreviation list of acronyms.
  • The caption of Figure 1 is too long. Rewrite.
  • Among the kind of carriers used to deliver vaccines, also niosomes have to be cited. For this purpose, see this recent work: Baldino & Reverchon, Niosomes formation using a continuous supercritical CO2 process, Journal of CO2 Utilization2021, 52, 101669; etc..
  • Some typing errors are present. Check and correct them.
  • Use the template of the Journal.

Reviewer 3 Report

In  Tables 2 and 3 the authors include a summary of DNA and RNA vaccines inoculated via respiratory tract.

These 2 tables could be more complete by including the main results of the examples included. 

A graphical abstract could also be included in the manuscript.

Reviewer 4 Report

The Review entitled Nanotechnologies in Delivery of DNA and mRNA Vaccines to the Nasal and Pulmonary Mucosa is very interesting and deal with an actual hot topic - DNA and RNA vaccines. The Authors discussed basic knowledge concerning this topic as well as presented interesting and important examples of such vaccines. The paper is in the scope of the journal however manuscript has to be rewritten. Generally, it looks like a conglomerate of small fragments of texts, there are some repetitions and a lot of typos. Additional schemes, figures, or pictures will benefit.

Author Response

With this letter, we submit our revised manuscript, titled “Nanotechnologies in Delivery of DNA and mRNA Vaccines to the Nasal and Pulmonary Mucosa” (Authors: Jie Tang, Larry Cai, Chuanfei Xu, Si Sun, Yuheng Liu, Joseph Rosenecker and Shan Guan) to Nanomaterials. We have extensively revised the manuscript (ID: nanomaterials-1465955) according to the reviewers’ comments. We think the revised manuscript has probably solved primary concerns raised by all the reviewers, to be more specific:

New content has been added in the revised manuscript, such as a list of acronyms/abbreviations, more information on the kind of nanotechnologies required for the loading and delivery of DNA and mRNA vaccines to the nasal and pulmonary mucosa in the abstract, publications on niosome-based carriers, main results of the cited examples in Tables 2 and 3, and a scheme figure summarizing current mRNA vaccines (Figure 2), as well as a graphical abstract.

We have also rewritten the manuscript in order to correct all the typos and errors in the previous submission, to our best efforts, along with cutting repetitions for more concise reading.

In addition, all the changes (marked in red) can be clearly found in the uploaded file named as “Revised manuscript 1465955 in template with Track Changes” as well as in “point by point responses to reviewers’ comments” attached below. We have also transferred all the content of the revised manuscript into the journal template. We also upload a file named as “Revised manuscript 1465955 in template clean version” as a clean version of the manuscript which would be easier for the reviewers to read.

We highly appreciate your help in reviewing our manuscript. If there is any question regarding the revised manuscript, please contact us at your convenience.

Yours sincerely,

Joseph Rosenecker

Round 2

Reviewer 1 Report

The review is interesting and only minor changes are required.

Line 74: clinical trials have been done with naked mRNA, for example Weide et al. J Immunother. Feb-Mar 2008;31(2):180-8

Line 85: Ref 20 is DC vaccine (transfected with mRNA). Better mention here more relevant mRNA vaccines such as Kranz et al. Nature. 2016 Jun 16;534(7607):396-401

Table 1: DNA vaccines is qualified as stable and RNA vaccine “relatively unstable”. However, it is the formulation that makes NA vaccines stable or not in vitro and thus it is similar for DNA (as mentioned line 352 and lines 466-469) or RNA vaccines: if the vehicle is unstable, the vaccine is unstable. DNA and RNA in nuclease free solutions are both stable.

Line 337: the 1990 publication by Wolff 1990 is not about immunization. Only protein expression. Line 339 “ It was thereafter demonstrated that…”

Line 365: A plasmid DNA vaccine against COVID-19 was approved in India. It could be mentioned here

Line 388 “determines”

Line 433 “chitosan on the are mucosal”?

Niosome is sometimes written noisomes (lines 475, 478, 483)

Line 499 “preferential uptake of DNA vaccines into DCs is desirable, since macrophages are not professional APCs..”. That is not correct. Macrophages are APC and vaccines getting to macrophages may work well.

Line 650: original publications for both Moderna and BioNTech/Pfizer mRNA vaccines would deserve to be mentioned here.

Line 672-673 reference about “novel adjuvants”?

Line 715: reference CleanCap: Henderson et al Curr Protoc. 2021 Feb;1(2):e39.

Line 716: not BNT162b1 but BNT162b2. Is Cleancap not used by Moderna?

Line 750: the first description of synthetic mRNA purification for HPLC is Pascolo Expert Opin Biol Ther. 2004 Aug;4(8):1285-94

Line 755: CureVac’s RNA vaccine does not use nucleotide modification and induces high antibody response (however low protection against COVID, possibly due to variants). Thus, modifications are not required for RNA vaccines: induction of strong immune responses by CureVac’s vaccine against COVID19 and strong immune responses by BioNTech’s vaccines against cancer that are also not modified

Author Response

For Reviewer #1:

Comments: The review is interesting and only minor changes are required.

Our response: We really appreciate Reviewer 1’s valuable comments for us to further improve the quality of this review and try to elucidate the following comments.

Comment 1: Line 74: clinical trials have been done with naked mRNA, for example Weide et al. J Immunother. Feb-Mar 2008;31(2):180-8

Our response: We thank Reviewer 1 for the suggestion. We have carefully studied and cited the highlighted example, but found that our existing example (citation 8, as shown below) already serves as a most recent case for the applicability of naked mRNAs in clinical settings.

8. Anttila, V.; Saraste, A.; Knuuti, J.; Jaakkola, P.; Hedman, M.; Svedlund, S.; Lagerström-Fermér, M.; Kjaer, M.; Jeppsson, A.; Gan, L.-M. Synthetic mRNA Encoding VEGF-A in Patients Undergoing Coronary Artery Bypass Grafting: Design of a Phase 2a Clinical Trial. Mol. Ther. Methods Clin. Dev. 2020, 18, 464–472.

Comment 2: Line 85: Ref 20 is DC vaccine (transfected with mRNA). Better mention here more relevant mRNA vaccines such as Kranz et al. Nature. 2016 Jun 16;534(7607):396-401.

Our response: We thank Reviewer 1 for the suggestion. We deleted the Ref 20 in the manuscript of previous version and cited the suggested example (Ref 23 in revised manuscript)

23. Kranz, L.M.; Diken, M.; Haas, H.; Kreiter, S.; Loquai, C.; Reuter, K.C.; Meng, M.; Fritz, D.; Vascotto, F.; Hefesha, H.; et al. Systemic RNA delivery to dendritic cells exploits antiviral defence for cancer immunotherapy. Nature 2016, 534, 396–401.

Comment 3: Table 1: DNA vaccines is qualified as stable and RNA vaccine “relatively unstable”. However, it is the formulation that makes NA vaccines stable or not in vitro and thus it is similar for DNA (as mentioned line 352 and lines 466-469) or RNA vaccines: if the vehicle is unstable, the vaccine is unstable. DNA and RNA in nuclease free solutions are both stable.

Our response: Thanks for Reviewer 1’s comment. We have included Reviewer 1’s opinion that the stability of DNA or RNA vaccines should depend on the formulations. Subsequently, DNA vaccines is qualified as “Ease of storage and transportation in most cases” and RNA vaccine as “Cold chain transportation is generally required” in Table 1 of the revised manuscript.

Comment 4: Line 337: the 1990 publication by Wolff 1990 is not about immunization. Only protein expression. Line 339 “ It was thereafter demonstrated that…”

Our response: Thanks to Reviewer 1 for pointing out the misleading description in Line 337 of previous manuscript. We have corrected it and rephrased the sentence as “The first proof of concept for in vivo protein expression with nucleic acids was reported in 1990 by injecting DNA or RNA molecules into mouse skeletal muscle for the expression of chloramphenicol acetyltransferase, luciferase, and galactosidase”. And the following sentence has been changed to “It was thereafter demonstrated that the production of cytotoxic T lymphocytes for influenza can be induced by injecting plasmid DNA (pDNA) encoding influenza A nucleoproteins into the quadriceps of BALB/c mice”.

Comment 5: Line 365: A plasmid DNA vaccine against COVID-19 was approved in India. It could be mentioned here

Our response: This is a very good suggestion and we have added the following discussion accordingly in the revised manuscript and cited the related references.

“Indeed, the first DNA vaccine to be approved for human use is a COVID-19 DNA vaccine (ZyCoV-D) developed in India. It was found to be 67% protective in clinical trials [103], providing evidence that DNA vaccines can be effective in controlling the pandemic [104]”

103. Sheridan, C. First COVID-19 DNA vaccine approved, others in hot pursuit. Nat. Biotechnol. 2021, 39, 1479–1482.

104. Mallapaty, S. India’s DNA COVID vaccine is a world first - more are coming. Nature 2021, 597, 161–162.

Comment 6: Line 388 “determines”

Our response: Thanks to Reviewer 1 for pointing out this error. We have corrected it and labelled with track changes in the revised manuscript.

Comment 7: Line 433 “chitosan on the are mucosal”?

Our response: We thank Reviewer 1 for pointing out this error. We have corrected this typing error. Amendments have been labeled with track changes in the revised manuscript.

Comment 8: Niosome is sometimes written noisomes (lines 475, 478, 483)

Our response: We thank Reviewer 1 for pointing out this typing error. We have corrected it and marked with track changes in the revised manuscript.

Comment 9: Line 499 “preferential uptake of DNA vaccines into DCs is desirable, since macrophages are not professional APCs..”. That is not correct. Macrophages are APC and vaccines getting to macrophages may work well.

Our response: We thank Reviewer 1 for this comment. We agree with Reviewer 1 that macrophages are APC and vaccines getting to macrophages may work well. We have deleted the misleading description “since macrophages are not professional APCs which effectively mediate adaptive immunity, but primarily clean the respiratory tract of particle debris.”, which marked with track changes in the revised manuscript.

Comment 10: Line 650: original publications for both Moderna and BioNTech/Pfizer mRNA vaccines would deserve to be mentioned here.

Our response: We thank Reviewer 1 for this suggestion. The original publications for both Moderna and BioNTech/Pfizer mRNA vaccines have been mentioned (i.e. Ref 25-27 in revised manuscript).

25. Jackson, L.A.; Anderson, E.J.; Rouphael, N.G.; Roberts, P.C.; Makhene, M.; Coler, R.N.; McCullough, M.P.; Chappell, J.D.; Denison, M.R.; Stevens, L.J.; et al. An mRNA Vaccine against SARS-CoV-2 - Preliminary Report. N. Engl. J. Med. 2020, 383, 1920-1931.

26. Baden, L.R.; El Sahly, H.M.; Essink, B.; Kotloff, K.; Frey, S.; Novak, R.; Diemert, D.; Spector, S.A.; Rouphael, N.; Creech, C.B.; et al. Efficacy and Safety of the mRNA-1273 SARS-CoV-2 Vaccine. N. Engl. J. Med. 2021, 384, 403-416.

27. Walsh, E.E.; Frenck, R.W.J.; Falsey, A.R.; Kitchin, N.; Absalon, J.; Gurtman, A.; Lockhart, S.; Neuzil, K.; Mulligan, M.J.; Bailey, R.; et al. Safety and Immunogenicity of Two RNA-Based Covid-19 Vaccine Candidates. N. Engl. J. Med. 2020, 383, 2439–2450.

Comment 11: Line 672-673 reference about “novel adjuvants”?

Our response: We thank Reviewer 1 for this reminder. We have added relevant references regarding “adjuvants” (i.e. Ref 164-166 in revised manuscript) as shown below.

164. Brito, L.A.; Chan, M.; Shaw, C.A.; Hekele, A.; Carsillo, T.; Schaefer, M.; Archer, J.; Seubert, A.; Otten, G.R.; Beard, C.W.; et al. A cationic nanoemulsion for the delivery of next-generation RNA vaccines. Mol. Ther. 2014, 22, 2118–29.

165. Li, Q.; Ren, J.; Liu, W.; Jiang, G.; Hu, R. CpG Oligodeoxynucleotide Developed to Activate Primate Immune Responses Promotes Antitumoral Effects in Combination with a Neoantigen-Based mRNA Cancer Vaccine. Drug Des. Devel. Ther. 2021, 15, 3953–3963.

166. Haabeth, O.A.W.; Lohmeyer, J.J.K.; Sallets, A.; Blake, T.R.; Sagiv-Barfi, I.; Czerwinski, D.K.; McCarthy, B.; Powell, A.E.; Wender, P.A.; Waymouth, R.M.; et al. An mRNA SARS-CoV-2 Vaccine Employing Charge-Altering Releasable Transporters with a TLR-9 Agonist Induces Neutralizing Antibodies and T Cell Memory. ACS Cent. Sci. 2021, 7, 1191–1204.

Besides, we think it is not accurate to use the word “novel”, since these adjuvants have been widely used for subunit vaccines previously. We, as a result, changed the “novel adjuvants” into “adjuvants proven to potentiate subunit vaccines” in the revised manuscript.

Comment 12: Line 715: reference CleanCap: Henderson et al Curr Protoc. 2021 Feb;1(2):e39.

Our response: We highly appreciate Reviewer 1 for this suggestion. We have added the suggested reference (i.e. Ref 173 in revised manuscript) as shown below.

173. Henderson, J.M.; Ujita, A.; Hill, E.; Yousif-Rosales, S.; Smith, C.; Ko, N.; McReynolds, T.; Cabral, C.R.; Escamilla-Powers, J.R.; Houston, M.E. Cap 1 Messenger RNA Synthesis with Co-transcriptional CleanCap(®) Analog by In Vitro Transcription. Curr. Protoc. 2021, 1, e39.

Comment 13: Line 716: not BNT162b1 but BNT162b2. Is Cleancap not used by Moderna?

Our response: Thanks to Reviewer 1 for pointing out these mistakes. We have corrected “BNT162b1” to “BNT162b2” and added “mRNA-1273” from Moderna in the revised manuscript with tracked changes.

Comment 14: Line 750: the first description of synthetic mRNA purification for HPLC is Pascolo Expert Opin Biol Ther. 2004 Aug;4(8):1285-94

Our response: We thank Reviewer 1 for this reminder. We have added the suggested publication (i.e. Ref 190 in revised manuscript) as shown below.

190. Pascolo, S. Messenger RNA-based vaccines. Expert Opin. Biol. Ther. 2004, 4, 1285–1294.

Comment 15: Line 755: CureVac’s RNA vaccine does not use nucleotide modification and induces high antibody response (however low protection against COVID, possibly due to variants). Thus, modifications are not required for RNA vaccines: induction of strong immune responses by CureVac’s vaccine against COVID19 and strong immune responses by BioNTech’s vaccines against cancer that are also not modified.

Our response: Thanks to Reviewer 1 for this comment. We agree with the reviewer that nucleotide modifications are not 100% necessary for IVT-mRNA vaccines. But the content in Line 755 of our previous manuscript is just a statement of fact that nucleotide modification technology has been employed in the particular production of BNT162b2 and mRNA-1273 COVID-19 mRNA vaccines. Besides, IVT-mRNA with nucleotide modifications indeed works much better compared to the unmodified counterpart in our hands, so we personally believe the nucleotide modification technology is very important and will lead the future development of IVT-mRNA based vaccines and therapeutics.

Reviewer 4 Report

Manuscript was improved

Author Response

For Reviewer #4:

Comments: Manuscript was improved.

Our response: We do appreciate Reviewer 4’s all valuable comments for us to improve the quality of this review.